# Detecting topology freezing transition temperature of vitrimers by AIE luminogens

Yang Yang [1], Shuai Zhang[1], Xiqi Zhang [2], Longcheng Gao[3], Yen Wei[1,4] & Yan Ji [1]

Vitrimers are one kind of covalently crosslinked polymers that can be reprocessed. Topology freezing transition temperature ($T_v$) is vitrimer's upper limit temperature for service and lower temperature for recycle. However, there has been no proper method to detect the intrinsic $T_v$ till now. Even worse, current testing methods may lead to a misunderstanding of vitrimers. Here we provide a sensitive and universal method by doping or swelling aggregation-induced-emission (AIE) luminogens into vitrimers. The fluorescence of AIE-luminogens changes dramatically below and over $T_v$, providing an accurate method to measure $T_v$ without the interference of external force. Moreover, according to this method, $T_v$ is independent of catalyst loading. The opposite idea has been kept for a long time. This method not only is helpful for the practical application of vitrimers so as to reduce white wastes, but also may facilitate deep understanding of vitrimers and further development of functional polymer materials.

[1] The Key Laboratory of Bioorganic Phosphorus Chemistry & Chemical Biology (Ministry of Education), Department of Chemistry, Tsinghua University, 100084 Beijing, China. [2] CAS Key Laboratory of Bio-inspired Materials and Interfacial Science, Technical Institute of Physics and Chemistry, Chinese Academy of Sciences, 100190 Beijing, China. [3] Key Laboratory of Bio-Inspired Smart Interfacial Science and Technology of Ministry of Education, School of Chemistry, Beihang University, 100191 Beijing, China. [4] Department of Chemistry, Center for Nanotechnology and Institute of Biomedical Technology, Chung-Yuan Christian University, 32023 Chung-Li, Taiwan, China. Correspondence and requests for materials should be addressed to L.G. (email: lcgao@buaa.edu.cn) or to Y.J. (email: jiyan@mail.tsinghua.edu.cn)

In 2011, Leibler et al. reported a radically new type of polymer—vitrimer[1]. Traditionally, polymers are divided into two classes: thermoplastics and thermosets. Thermosets are covalently crosslinked networks. They play substantial roles in aircrafts, vehicles, buildings, electronics, and so on, as they are chemical-resistant and extremely durable. However, thermosets have a big disadvantage: unlike thermoplastics, they cannot be reprocessed due to their insoluble and infusible nature. They are not recyclable and contribute to huge plastic wastes on our planet[2,3]. Vitrimers ignite the hope to settle this problem. Even though vitrimers are covalently crosslinked, they can be deformed like silica glass when heated, enabling vitrimers to be self-healable, weldable, remouldable, reshapable, and thus repeatedly recyclable[1,4–12]. The reprocessability of vitrimers comes from rapid topology changes due to stimuli-triggered exchangeable reactions. During the topology change, the number of crosslinks is constant. In the past 8 years, various vitrimers have been created[1,4–6,8,9,13–17].

It is vital to know vitrimers' upper limit temperature for use and lower limit temperature for recycle. For common polymers, they have their characteristic temperatures such as glass transition temperature ($T_g$), melting point ($T_m$), and so on. Those temperatures provide an indication on the safe utilization temperature. For vitrimers, there is another temperature, called topology freezing transition temperature ($T_v$)[1,11,18], which is also very important. Above $T_v$, exchangeable reaction happens fast and vitrimer is able to be reprocessed and recycled; below $T_v$, exchangeable reaction is slow and vitrimer is similar to the traditional thermoset. $T_v$ not only determines the upper limit temperature for vitrimer to use, but also has a direct impact on the vitrimer's performance and reprocessing.

However, it is hard to detect the intrinsic $T_v$. It is generally regarded that $T_v$ resembles $T_g$[1,11,18]. This is because the viscosity decrease of vitrimer follows the Arrhenius law from elastic phase to viscoelastic liquid state as temperature increases, just like the viscosity change of silica during the $T_g$ range[1,18]. The exchangeable reaction rates below and above $T_v$ are different. As the exchangeable reaction happens in the network, it is hard to find $T_v$ by monitoring the reaction kinetics. At present, dilatometry test[1,19,20] and stress-relaxation test measured by rheology[1,15,18,21–23] or dynamic mechanical analyzer (DMA)[24] are used to measure $T_v$. In both methods, the measurements are done while the samples are under external force. It has been proved that an additional local force provides an extra tension on the crosslinking bonds, which affects the breakage rate of crosslinks and effective activation energy. Subsequently, external force may induce a shift in $T_v$[24,25]. Experimental parameters strongly affect the results. In other words, so far, there are no existing methods that can reflect the actual $T_v$ in static-situation, which is closely related to the intrinsic nature of the network. Such situation brings troubles to the research and practical application of vitrimers. For a typical example, according to our experience, soft actuators made of certain liquid crystalline vitrimers[5] are supposed to be stable when the utilization temperature is under $T_v$, which is about 160 °C measured by dilatometry, but they lose actuation quickly in static condition even when the temperature is 40 °C lower than $T_v$. It is dangerous to use a material when its upper service temperature is unknown. Therefore, it is highly demanded to find a proper accurate method to measure the true $T_v$, which is similar to differential scanning calorimetry (DSC) used to measure the upper utilization temperature of traditional polymers (e.g., $T_g$ or $T_m$).

In this paper, we put forward a simple method to measure $T_v$ under a static state using aggregation-induced-emission (AIE) luminogens as fluorescent probes. AIE molecules are organic compounds with excellent emission properties in aggregated state

or in solid state, which was discovered by Tang and co-workers[26,27]. The application of AIE phenomenon has been extended to various areas, such as electroluminescence devices, fluorescent sensors, cell imaging, and so on[26–28]. The aggregation of AIE molecule increases the restriction of intramolecular motions (RIM) and results in intense fluorescence emission[29,30]. Tang and co-workers adopted AIE luminogens as probes into polymers to determine their $T_g$[31–33]. When temperature rises, polymer's fluorescent intensity decreases. While during $T_g$ range, the decreasing rate varies. So $T_g$ can be determined by this turning point of decreasing rate. This AIE-probe technique is very accurate, straightforward, and reliable. Inspired by their work, we suppose that AIE-probe technique may be used to measure $T_v$ based on following possible mechanisms. Below $T_v$, vitrimer is a vitrified crosslinked network. AIE molecule should be restricted to intramolecular motion. The energy of the excited state will decay and be annihilated by radiation, thus AIE will be highly emissive in this state. Above $T_v$, the network rearranges due to the accelerated exchangeable reaction. The movement of the network will greatly increase the freedom for AIE to motion intramolecularly. The activated intramolecular rotations will efficiently deactivate its excitons non-radiatively and serve as a nonradiative channel to decay the energy of the excited state, and then weaken the fluorescence emission of AIE (Fig. 1). If this hypothesis is correct, we should be able to observe a change around $T_v$ in the fluorescence intensity plot. Here in this paper, we prove that the above hypothesis is correct and the AIE-probe technique can be used to measure $T_v$ of vitrimers.

## Results

**Detecting $T_v$ of epoxy vitrimers by AIE luminogens.** We use a typical epoxy vitrimer to testify the validity of this method. The epoxy vitrimer was synthesized by reacting equimolar diglycidyl ether of bisphenol-A with adipic acid and using 1,5,7-triazobicyclodecen (TBD, 5 mol% to the COOH) as a catalyst (Fig. 2a). Commercially available tetraphenylethene (TPE) (1 wt%) was doped as a fluorescent probe[34,35]. The sample looks the same as the neat epoxy vitrimer without TPE. Upon UV irradiation (365 nm), TPE-doped epoxy vitrimer emits strong cyan light (~470 nm) at room temperature (Supplementary Fig. 4a). As the neat epoxy vitrimer emits extremely weak light at ~425 nm (Supplementary Fig. 3), the fluorescence of the blank sample has no significant effect. From X-ray diffractometry (XRD) curves in Supplementary Fig. 2a, we can deduce that TPE is homogenously

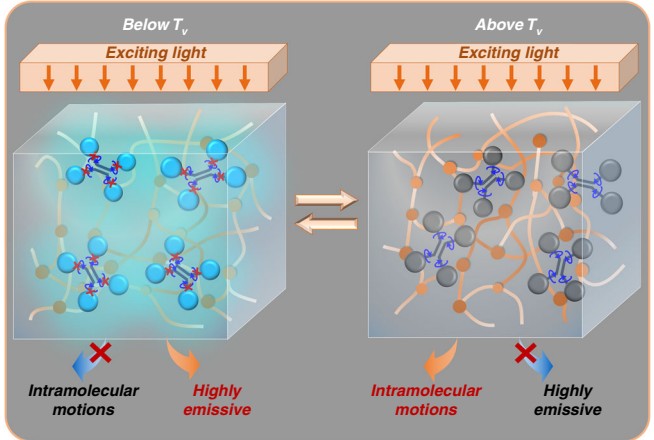

**Fig. 1** AIE-doped vitrimer is highly emissive below $T_v$ and weakly emissive above $T_v$. Below $T_v$, AIE molecule is restricted to intramolecular motion. While above $T_v$, the activated intramolecular rotations of AIE decay the energy of the excited state, and then weaken its fluorescence emission

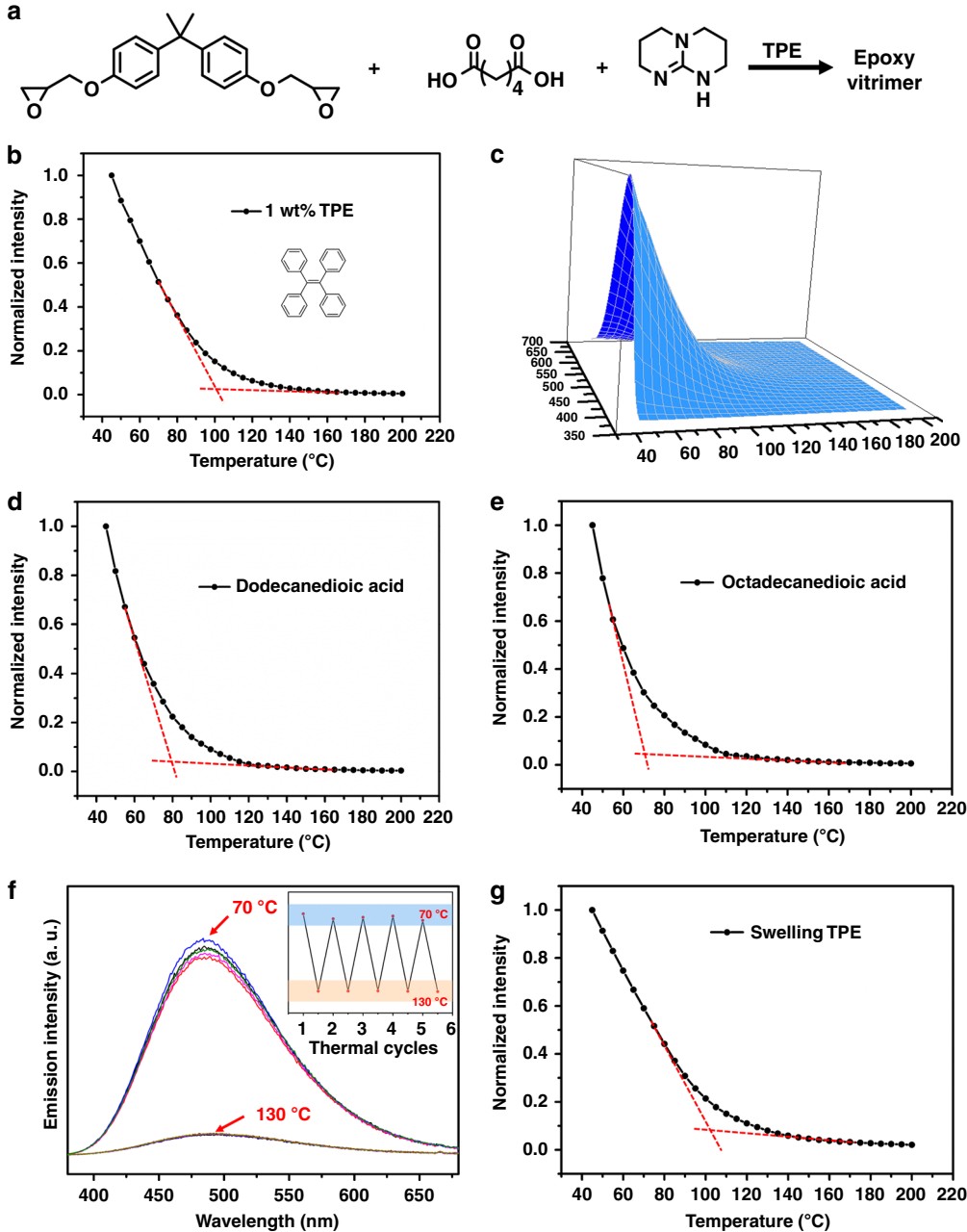

**Fig. 2** Fluorescent properties of TPE-doped epoxy vitrimer. **a** Synthesis of TPE-doped epoxy vitrimer. **b** The fluorescence intensity plot of 1 wt% TPE-doped epoxy vitrimer at a series of temperature at 470 nm (Excitation: 365 nm). All the fluorescence intensities are normalized to 45 °C. **c** Fluorescence spectra of 1 wt% TPE-doped epoxy vitrimer at a series of rising temperatures. Excitation: 365 nm. **d** The fluorescence intensity plot of 1 wt% TPE-doped epoxy vitrimer cured by dodecanedioic acid at a series of temperature at 470 nm (Excitation: 365 nm). All the fluorescence intensities are normalized to 45 °C. **e** The fluorescence intensity plot of 1 wt% TPE-doped epoxy vitrimer cured by octadecanedioic acid at a series of temperature at 470 nm (Excitation: 365 nm). All the fluorescence intensities are normalized to 45 °C. **f** Fluorescence spectra of 1 wt% TPE-doped epoxy vitrimer at 70 °C and 130 °C by switching several cycles. Excitation: 365 nm. **g** The fluorescence intensity plot of TPE-swelled epoxy vitrimer at a series of temperature at 470 nm (Excitation: 365 nm). All the fluorescence intensities are normalized to 45 °C

dispersed in the polymer matrix. What is more, doping 1 wt% TPE does not obviously affect the thermal and mechanic properties of the epoxy vitrimer. There are no evident differences between the neat epoxy vitrimer and that doped with 1 wt% TPE in terms of $T_g$ (about 45 °C) detected by DSC (Supplementary Fig. 1a), mechanical strength tested by DMA (Supplementary Fig. 1b) and thermal stability measured by thermal gravity analysis (TGA, Supplementary Fig. 1c). Deserved to be mentioned, $T_v$ cannot be distinguished from DSC curves (Supplementary Fig. 1a).

Indeed, the transition and the turning point of fluorescence intensity can be clearly identified. We measured the fluorescence spectrums (excitation wavelength: 365 nm) of 1 wt% TPE-doped epoxy vitrimer at a serial of temperatures from 45 °C to 200 °C (well below its decomposition temperature, Supplementary Fig. 1c) with the temperature interval of 5 °C. As the temperature of the heating apparatus is unstable below 45 °C, we started from 45 °C. Normalized fluorescence intensity plot at the wavelength of 470 nm is shown in Fig. 2b. The fluorescence intensities at different temperatures are normalized to 45 °C. As temperature

increases, fluorescence intensity decreases. We observed a transition between 85 and 150 °C and a turning point at about 102 °C (the intersection of two tangent lines, Fig. 2b, c), which indicate $T_v$ range and $T_v$ value, respectively. $T_v$ value detected here (102 °C) is pretty different to our previous result measured by dilatometry test[12]. It is well known that intrinsic fluorescence intensity of TPE powders also decreases as temperature rises. When AIE-probe technique was used to determine $T_g$ of polymers (such as polystyrene (PS, $T_g$ is 80 °C), and poly(methyl methacrylate) (PMMA, $T_g$ is 103 °C))[31], there was just one turning point of $T_g$ in the fluorescence intensity plots if AIE is homogeneously dispersed. That is to say, there is no turning point derived from AIE itself. Therefore, here the transition is not caused by TPE itself, but by the appearance of $T_v$.

**The merits of this AIE-probe technique and further understanding of the vitrimers**. There are several notable merits of this method. First of all, this method has a very good sensitivity. From Supplementary Fig. 5, we can see that even quite a few content of TPE (0.1 wt%) can be used to detect $T_v$ with a high sensitivity and accuracy. This means that $T_v$ can be reliably and easily measured in spite of the little amount of TPE loading. It is also possible to improve the fluorescent sensitivity by simply increasing the probe loading. As this method is very sensitive, small variation on the chemical structure of vitrimers can be clearly found out. As shown in Fig. 2d, e, when epoxy vitrimers are cured by dodecanedioic acid and octadecanedioic acid, the $T_v$ is ~82 °C and ~70 °C, respectively. The longer alkyl chain, the lower $T_v$. This difference is hard to be distinguished by dilatometry tests (Supplementary Fig. 7). Secondly, this method has a very good reproducibility. As shown in Fig. 2f, after switching the temperature between 70 °C (below $T_v$) and 130 °C (above $T_v$) for several cycles, the fluorescence intensity is almost the same at both temperatures. Thirdly, compared to previous methods[1], which showed that $T_v$ was heating-rate dependent, this method is independent to heating rate, as just the fluorescence intensity at a certain temperature is needed, regardless of what heating rate is used to reach that temperature. Fourthly, it does not matter whether the AIE luminogens are incorporated into vitrimers during synthesis or introduced into the cured vitrimer after the synthesis by swelling. As a demonstration, TPE was added into the epoxy vitrimer by immersing a piece of epoxy vitrimer into the TPE chloroform solution (concentration: 16.46 mg/mL) for 2 hours. As shown in (Fig. 2g), the turning point of $T_v$ is still 102 °C. That is to say, to use this method, it is not necessary to add AIE luminogens at the beginning of vitrimer synthesis. Therefore, there is no need to synthesis on purpose vitrimer with AIE luminogens in order to know its $T_v$. $T_v$ of the existing vitrimers can also been measured by swelling AIE molecule into them. Fifthly, a wide selection to AIE molecules can be used. Besides TPE, we also tried 1,1,2,3,4,5-hexaphenylsilole (HPS)[30] and 1,2,3,4-tetraphenyl-1,3-cyclopentadiene (TPCP)[26], the structures of which are shown in Supplementary Fig. 6. From the result in Supplementary Fig. 6, we can see that for samples with 1 wt% TPCP and 0.1 wt% HPS respectively, $T_v$ can both be detected. This allows us to choose suitable AIE to avoid the florescence interference of vitrimers themselves when necessary.

The fluorescence intensity plots provide valuable information on the practical utilization of vitrimers. Figure 2b shows that $T_v$ is a temperature range, just like $T_g$, instead of a certain temperature. By previous methods including dilatometry test and stress-relaxation test measured by rheology or DMA, $T_v$ can only be identified or calculated as a certain temperature[1,8,12,18]. For example, by dilatometry test, $T_v$ of this epoxy vitrimer is about 160 °C[12]. From Fig. 2b, we can see that fluorescence intensity at

470 nm decreases linearly and drastically from 45 to 80 °C, while relatively slowly and nonlinearly between 85 and 150 °C, thereafter, linearly and slightly above 150 °C. Therefore, the possible topology change due to transesterification may start from 85 °C. $T_v$ is a range from 85 to 150 °C. For safety reasons, this vitrimer should be used at a temperature at least below 85 °C. We used a creep experiment to illustrate it. As shown in Fig. 3a, a binder clip (139.5 kPa) was attached to the samples as a load. After 10 min, the length of vitrimer strips increased 0, 0, 5, 6, 9, and 12.5% at 25C, 50C, 80, 100, 130, and 160 °C, respectively. Below 85 °C, creep is not obvious. For the temperatures between 85 and 150 °C, creep occurs at different extents. As indicted by the plot (Fig. 2b), the exchange reaction rate at 130 °C should be similar to that at 160 °C but different from that at 100 °C. This is consisted with our observation of the creep experiment.

Thanks to this method, we are able to verify that $T_v$ of the vitrimer is in fact not influenced by the catalyst content. According to previous studies, $T_v$ decreases as catalyst loading increases when measured by other techniques[18,36]. Our result here is totally different. As shown in Fig. 3b, as TBD loading increases from 0 to 7.5 mol% (to –COOH), $T_v$ of 1 wt% TPE-doped epoxy vitrimer is almost the same. As $T_v$ is detected in an absolutely static state, $T_v$ here reflects an intrinsic nature of the network, and has nothing to do with the catalyst. It is in accordance with the theory that $T_v$ should be an intrinsic feature of a vitrimer. Leibler and other researchers had confirmed that vitrimer's viscosity, which reflects exchangeable reaction, follows Arrhenius law (equation 1), and $E_a$ is unchanged when catalyst loading changes. The $k$ is small below $T_v$ and large above $T_v$. So, from its differential form in equation (2), the exchangeable reaction rate or viscosity ($\frac{d\ln k}{dT}$), which can be negatively reflected here by the fluorescence intensity plot as a function of increasing temperature precisely, is only connected with temperature, as $E_a$ is unchanged and only $T$ is a variable in $\frac{E_a}{RT^2}$. That is to say, $T_v$, signifying the exchangeable reaction rate ($\frac{d\ln k}{dT}$), is only connected with temperature and is not related to catalyst loading. Even though adding catalyst does change the pre-exponential factor $A$, catalyst accelerates the reaction at all temperatures. When taking a derivative with respect to temperature, $d(\ln A)$ is a zero. It does not affect the change of slope rate (which is used for identifying $T_v$) in fluorescence intensity-temperature plot.

$$k = Ae^{\frac{-E_a}{RT}} \tag{1}$$

$$\frac{d\ln k}{dT} = \frac{E_a}{RT^2} \tag{2}$$

Where $k$ is rate constant; $E_a$ is the activation energy for the reaction; $T$ is the absolute temperature (in kelvin); $R$ is the universal gas constant; $A$ is the pre-exponential factor, a constant for each chemical reaction.

**General applicability of the detecting methods**. To show this is a general method to detect $T_v$ of various vitrimers or other dynamic crosslinked networks with exchangeable bonds, we here use polyurethane and polyimine to illustrate. Recently, vitrimer features of polyurethanes, including remoulding, reconfiguring, and self-healing, have been developed based on transcarbamoylation (exchangeable reaction between essential carbamate bonds)[9,13]. As shown in Fig. 4a, we prepared a polyurethane by reacting glycerine (GLY), poly(ethyleneglycol)diol (PEG 400) with hexamethylene diisocyanate (HDI) in the presence of dibutyltin dilaurate (DBTDL, 0.5 wt%) as a catalyst, according to the method of Xie and co-workers[9]. As the blank polyurethane emits weak blue light (Supplementary Fig. 8a), here we doped 2,3-bis[4 (diphenylamino)phenyl]fumaronitrile (TPAFN)[37,38] (0.1 wt%),

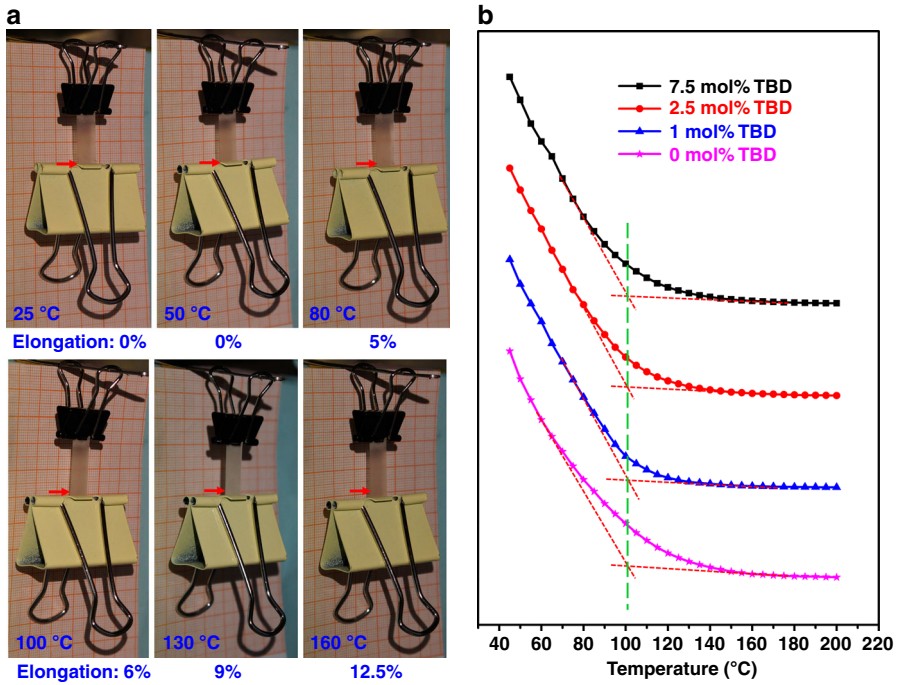

**Fig. 3** The thermo-behavior and fluorescence intensity plots with different catalyst loading of epoxy vitrimers. **a** Creep experiment of epoxy vitrimer at a serial of temperatures (25, 50, 80, 100, 130, and 160 °C) for 10 min, respectively. The length of them increased 0, 0, 5, 6, 9, and 12.5% at 25, 50, 80, 100, 130, and 160 °C. respectively. **b** The fluorescence intensity plots of 1 wt% TPE-doped epoxy vitrimer with different TBD catalyst loadings (7.5, 2.5, 1, and 0 mol% to the COOH) at a series of temperature at 470 nm (Excitation: 365 nm). All the fluorescence intensities are normalized to 45 °C

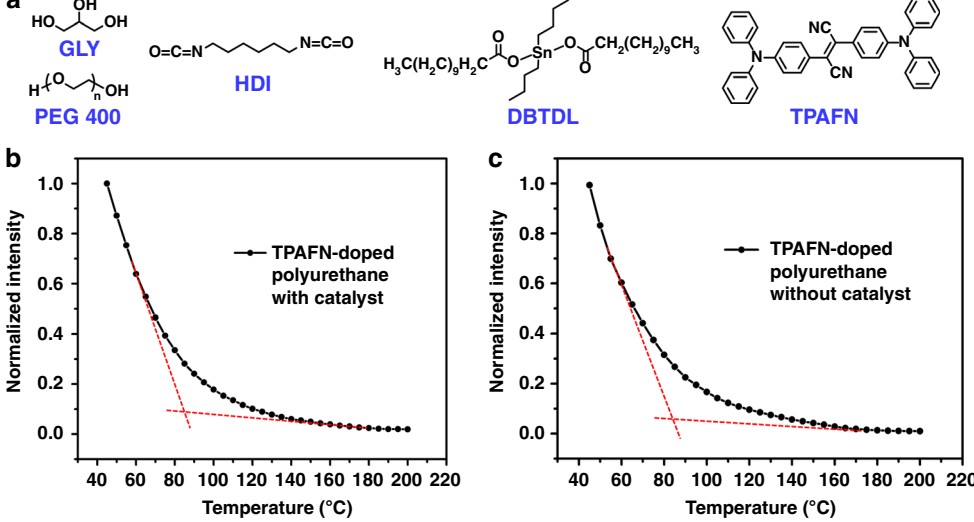

**Fig. 4** Detecting $T_v$ of polyurethanes by AIE-probe technique. **a** Synthesis of polyurethane. **b** The fluorescence intensity plot of 0.1 wt% TPAFN-doped polyurethane at a series of temperature at 633 nm (Excitation: 527 nm). All the fluorescence intensities are normalized to 45 °C. **c** The fluorescence intensity plot of 0.1 wt% TPAFN-doped polyurethane without DBTDL catalyst at a series of temperature at 633 nm (Excitation: 527 nm). All the fluorescence intensities are normalized to 45 °C

which emits red light as AIE probes to avoid the interference of intrinsic polyurethane fluorescence. According to DSC curves (Supplementary Fig. 8b), the $T_g$ of both TPAFN-doped polyurethane and blank polyurethane is about −20 °C. Normalized fluorescence intensity plot (excitation: 527 nm) at the wavelength of 633 nm of 0.1 wt% TPAFN-doped polyurethane is shown in Fig. 4b. We can see that $T_v$ is about 85 °C. From Fig. 4c, it can also be proved that catalyst loading does not affect $T_v$, the $T_v$ of

TPAFN-doped polyurethane without catalyst (DBTDL) remains at about 85 °C.

We also tested the validity of this method using polyimines. It had been reported that polyimines are malleable and recyclable because of the transamination[6]. The polyimime-1 used here is synthesized by commercially available ingredient: terephthaldehyde (TPA), 3,3′-diamino-N-methyldipropylamine (DMDPA), and trimethylolpropane tris[poly(propylene glycol), amine terminated]

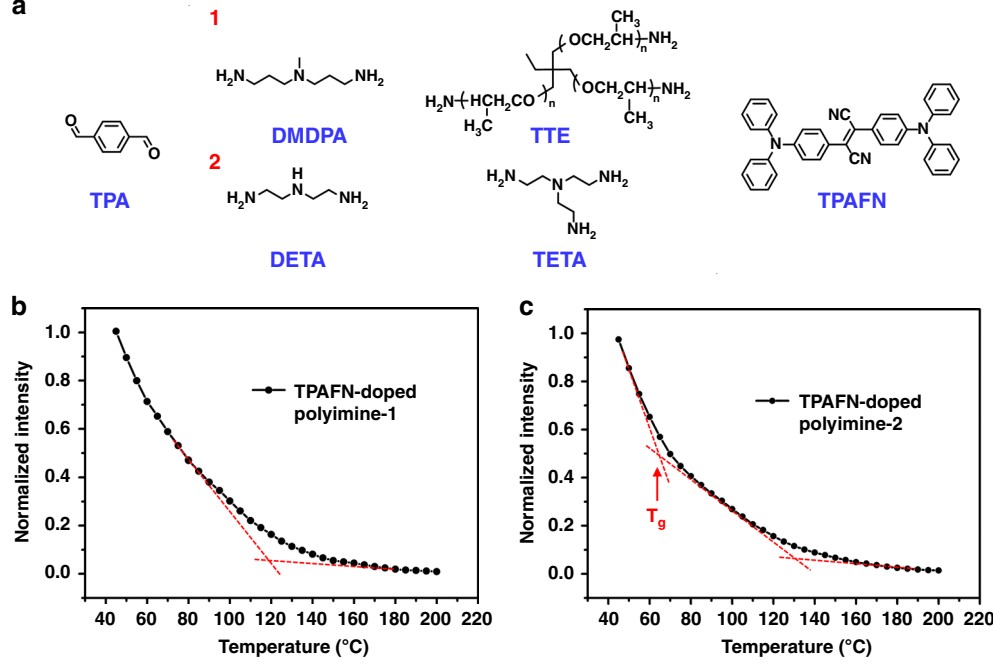

**Fig. 5** Detecting $T_v$ of polyimines by AIE-probe technique. **a** Synthesis of polyimine-1 and polyimine-2. **b** The fluorescence intensity plot of 0.5 wt% TPAFN-doped polyimine-1 at a series of temperature at 621 nm (Excitation: 527 nm). All the fluorescence intensities are normalized to 45 °C. **c** The fluorescence intensity plot of 0.5 wt% TPAFN-doped polyimine-2 at a series of temperature at 644 nm (Excitation: 527 nm). All the fluorescence intensities are normalized to 45 °C. The point at about 63 °C is $T_g$

ether (TTE) (Fig. 5a), according to the method of Zhang and co-workers[6]. As the blank polyimine-1 emits weak brown light (Supplementary Fig. 9a), we doped TPAFN (0.5 wt%) as AIE probes to avoid the effect of intrinsic polyimine-1 fluorescence. According to DSC curves (Supplementary Fig. 9b), the $T_g$ of both TPAFN-doped polyimine-1 and blank polyimine-1 is about 20 °C. To determine $T_v$ of polyimine-1, we measured the fluorescence spectrums (excitation: 527 nm) of TPAFN-doped polyimine-1 at a serial of temperatures from 45 to 200 °C with the temperature interval of 5 °C. Normalized fluorescence intensity plot at the wavelength of 621 nm of TPAFN-doped polyimine-1 is shown in Fig. 5b. We can see that $T_v$ is about 120 °C. When the components of the polyimine change, $T_v$ changes accordingly. We prepared another kind of polyimine-2 which was synthesized by terephthaldehyde (TPA), triethylene tetramine (TETA), and diethylene triamine (DETA) (Fig. 5a). The $T_v$ of polyimine-2 is about 130 °C (Fig. 5c), which confirms our conclusion above as well. It is worth mentioning that $T_g$ can also be detected by this method. We can see that its $T_g$ is about 63 °C, which corresponds to DSC curves (Supplementary Fig. 10b).

## Discussion

Vitrimer opens up great opportunities for science and industry. Since the emergence of vitrimer, how to detect $T_v$ has been a key factor that restricts its research and application. Like using DSC to detect $T_g$ or other transition temperatures, AIE-probes fluorescence technique here determines $T_v$ in a totally static state, which can reflect the intrinsic characteristics of the network. This method provides valuable information on the intrinsic feature of vitrimer network as temperature changes. Compared to previous methods that measure $T_v$ under external forces, this method is fast, simple, and with good repeatability. This technique excludes the influence of external force, which makes the results of different researchers comparable. As we have shown, $T_v$ is independent on the catalyst. The samples without catalyst used here are in fact traditional thermosets. This may also give us a clue on

whether a thermoset might be changed into a reprocessable malleable vitrimer by adding catalyst to facilitate the exchangeable reaction, which is already existed in the thermoset. Although we measure $T_v$ in a static state here, this method can also be used to monitor the influence of external force on $T_v$ under stress if sample holder is slightly modified.

## Methods

**Chemicals.** Triazobicyclodecene (TCI, 98%), adipic acid (J&K Scientific Ltd., 99%), octadecanedioic acid (SHUYA Chemical Science and Technology, 98%), dodecanedioic acid (Aladdin, 99%), diglycidyl ether of bisphenol A (Sigma-aldrich, D.E.R. 332), tetraphenylethene (TPE, Tianjin Heowns Biochemical Technology Co. Ltd., 98%), 1,1,2,3,4,5-hexaphenylsilole (HPS, TCI, > 98.0%), 1,2,3,4-tetraphenyl-1,3-cyclopentadiene (TPCP, J&K Scientific Ltd., 98%), glycerol (GLY, Biotopped Science & technology CO., Ltd.), polyethylene glycol (PEG-400, TCI), 1,6-diisocyanatohexane (HDI, J&K Scientific Ltd., 99%), dibutyltin dilaurate (DBTDL, Damasbeta, 95%), diethylenetriamine (J&K Scientific Ltd., 99%), p-phthalaldehyde (Aladdin, 98%), tris(2-aminoethyl)amine (Alfa Aesar, 97%), 3,3'-diamino-N-methyldipropylamine (Tianjin Heowns Biochemical Technology Co. Ltd., 98%) and trimethylolpropane tris[poly(propylene glycol), amine terminated] ether (Sigma-aldrich, Mn: 440) were used directly without further purification. 2,3-bis[4(diphenylamino)phenyl]fumaronitrile (TPAFN) was prepared according to the method described by Tang and co-authors (*Sci. Rep.-UK* 2016, 3, 1150).

**Preparation of AIE-doped epoxy vitrimers.** Stoichiometric amounts of diglycidyl ether of bisphenol A (1 mmol), desired diacid (1 mmol) and desired AIE loading were mixed and heated to 180 °C. After the mixture was melted, desired triazobicyclodecene loading (x mol% to the COOH groups) was introduced and stirred manually till homogeneous. As the mixture became very viscous, it was cooled to room temperature to obtain a solid product, which was not completely crosslinked. Then the solid was sandwiched between two plates to be cured by a hot press for 4 h at 180 °C. A spacer was placed between two plates to control the thickness of film. The applied pressure was 3 MPa. Fourier transform infrared spectroscopy (FTIR, *Perkin Elmer spectrum 100*) was used to monitor the reaction progress. The epoxy peak at 912 cm$^{-1}$ totally disappeared after curing for 4 h, indicating the complete reaction.

**Preparation of AIE-doped polyurethanes.** The polyurethanes were prepared according to the method described by Xie and co-authors (*Angew. Chem. Int. Ed.* 2016, 55, 11421). 0.5 g PEG-400, 0.115 g GLY (molar ratio of PEG:GLY = 1:1) and 0.1 wt% TPAFN were mixed in plastic centrifuge tube and sonicated for 1 h to be

more uniform. Then the mixture was moved into ice water and stirred. 0.525 g HDI and 0.5 wt% DBTDL catalyst were added into the tube and stirred for 30 min. The mixture was heated to 60 °C for 2 h. Finally, the solid was sandwiched between two plates to be cured by a hot press for 2 h at 140 °C. A spacer was placed between two plates to control the thickness of film. The applied pressure was 3 MPa. The TPAFN-doped polyurethane without DBTDL catalyst and blank polyurethane were prepared by above method as well. The difference is that there is no DBTDL catalyst and no TPAFN molecular in the first step, respectively.

**Preparation of AIE-doped polyimine-1**. The polyimines were prepared according to the method described by Zhang and co-authors (*Adv. Mater.* 2014, 26, 3938). Firstly, 1 molar equivalent of terephthaldehyde and 0.5 wt% TPAFN were dissolved in a minimum amount of dichloromethane:ethyl acetate:ethanol (1:1:8 volume ratio). Then 0.572 molar equivalents of 3,3′-diamino-N-methyldipropylamine and 0.286 molar equivalents of trimethylolpropane tris[poly(propylene glycol), amine terminated] ether were dissolved together in ethanol, and then added to a glass tray covered by PTFE tape before use. The terephthaldehyde solution was then added to the same tray. The solvent was allowed to evaporate in a fume hood under ambient conditions. The obtained solid was then gradually cured in a drying oven: 30 min at 75 °C, then 30 min at 85 °C, and finally 30 min at 105 °C. The solid was sandwiched between two plates to be cured by a hot press for 2 h at 140 °C. Blank polyimine-1 was prepared as above procedure. The difference is that there is no TPAFN molecular in the first step.

**Synthesis of TPAFN-doped polyimine-2**. Firstly, 1 molar equivalent of terephthaldehyde and 0.5 wt% TPAFN were dissolved in a minimum amount of dichloromethane:ethyl acetate:ethanol (1:1:8 volume ratio). Then 0.3 molar equivalents of diethylene triamine and 0.4667 molar equivalents of triethylene tetramine were dissolved together in ethanol, and then added to a glass tray covered by PTFE tape before use. The terephthaldehyde solution was then added to the same tray. The solvent was allowed to evaporate in a fume hood under ambient conditions. The obtained solid was then gradually cured in a drying oven: 30 min at 75 °C, then 30 min at 85 °C, and finally 30 min at 105 °C. The solid was sandwiched between two plates to be cured by a hot press for 2 h at 140 °C. Blank polyimine-2 was prepared as above procedure. The difference is that there is no TPAFN molecular in the first step.

**Thermal, mechanical, and XRD characterizations of AIE-doped epoxy vitrimers**. $T_g$ was measured by DSC (*TA-Q2000*) at a scanning rate of 5 °C /min. Tensile test was performed on a DMA (*TA-Q800*) apparatus in the tension film geometry under the controlled force mode at 30 °C, with a rectangular tension film dimension of $10.0 \times 2.5 \times 0.15$ mm and a ramp force of 0.1 N/min. The thermal stability was measured by TGA (*TA-Q50*) under air atmosphere at a heating rate of 20 °C /min. XRD measurements were performed using an XRD diffractometer (*Bruker*, *D8 ADVANCE*), having a wavelength of 0.154 nm. The diffractometer was scanned in the 2θ range from 5 to 70°, and the scanning rate used was 0.1 s/step.

**Fluorescence spectra of vitrimers**. All the fluorescence spectra tests were recorded using a steady state spectrometer (*NanoLog* infrared fluorescence spectrometer, *Nanolog FL3-2iHR*) equipped with a temperature control system. Fluorescence spectra were scanned every 5 °C from 45 °C to 200 °C for all specimens.

## Data availability
The authors declare that most data supporting the findings of this study are available within the paper and its Supplementary Information Files. The rest of the data are available from the corresponding author upon reasonable request.

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

## Acknowledgements

We thank Prof. Xi Zhang, Prof. Jiangfei Xu, and Bo Qin from Department of Chemistry, Tsinghua University for the help with DSC tests; Prof. Xiaoyong Zhang, Ruming Jiang, and Liucheng Mao from Department of Chemistry, Nanchang University for the help with synthesizing TPAFN; Dr. Yan Guan from Analytical Instrumentation Center, College of Chemistry and Molecular Engineering, Peking University for the help with the fluorescence spectrometer tests and analysis; Prof. Lei Tao, Dr. Guoqiang Liu, Tengfei Mao, and Yuan Zeng from Department of Chemistry, Tsinghua University for the help with figures and analysis. This research was supported by the National Natural Science Foundation of China (Nos. 51722303, 21674057, 21805159, 21788102, and 21875009) and China Postdoctoral Science Foundation (Nos. 2018T110086 and 2017M620735).

## Author contributions

Y.J. and Y.Y. proposed and designed the project. L.G. participated in the project. Y.Y. performed all the experiments with the help of S.Z., X.Z., and Y.W.; Y.Y., Y.J., and L.G. wrote the paper together. All authors discussed the results and commented on the manuscript.

## Additional information

**Competing interests:** The authors declare no competing interests.

