## [Transparent Peer Review File · Nature Communications]

Reviewers' comments:

Reviewer #1 (Remarks to the Author):

This paper describes a new ingenious way to determine T_v in vitrimers. Vitrimers are the subject of intense research both in basic science but also for practical applications. However, one of the limits in the application of vitrimers in real world, is the accurate measurement of the "working temperature" for them.

T_v , topology freezing transition temperature, represents the temperature limit above which the vitrimers cannot be used anymore for practical applications.

Authors propose to use AIE molecules as internal probes to determine T_v . As the fluorescence intensity of AIE molecules is sensitive to its environment (high intensity when the internal rotations of AIE are restricted, lower intensity when more freedom around AIE molecules increases the internal rotations of the AIE molecules), authors speculate that there will be possible to detect T_v by following the evolution of the fluorescence intensity with the T variations.

The technique works perfectly well. Authors tested the technique on different vitrimers to demonstrate the "universality" of the technique.

This paper, which describes a very original technique toward the accurate measurements of T_v in vitrimers, will have a great impact in this domain of research, which is a very hot subject. This paper is suitable for publication in Nature Communications. However, the English needs to be improved substantially.

P. Keller

Reviewer #2 (Remarks to the Author):

This paper reported a sensitive fluorescence method for detection of topology freezing transition temperature (T_v). Luminogens with aggregation-induced-emission (AIE) characteristics were used as probes. The fluorescence intensity of polymer vitrimers doped with AIE luminogens changed sharply below and over T_v , which allowed detection of T_v . Moreover, It was found that T_v was independent of catalyst loading, by using the detection method. The method reported is novel and interesting. Questions are listed as follows:

1. How does aggregation of dyes affect detection of T_v ? In epoxy vitrimers, TPE is homogeneously dispersed in polymer matrix. If dyes form aggregates in polymer matrices, does the method work well? Does inhomogeneous dispersion of dyes affect the results?
2. The authors chose the polymer vitrimers that are compatible with dyes. For the incompatible systems, does the method work well?

In summary, I recommend its publication after minor revision.

Reviewer #3 (Remarks to the Author):

The authors apply the well-known effect of AIE to detect a change in fluorescence in doped vitrimers.

The idea is ingenious and novel but the data interpretation in terms of topology-freezing transition is not sufficiently discussed in the paper, containing some serious methodological approximations that need to be better justified.

- 1) A key point is to understand where does the change in fluorescence come from and what is the meaning of T_v based on what is known about AIE. Figure 1 is completely unclear. What difference should the reader see between above and below T_v ? Where is the decrease in intensity coming

from in a solid epoxy ? Are the luminogens becoming aggregated, i.e. insoluble in the epoxy. Is T_v the temperature where they become insoluble ? If the luminogens can be infused in the epoxy with a solvent they are presumably mobile (the epoxy is above its T_g). Would the decrease in intensity not be due to a thermodynamically driven phase separation between luminogens and epoxy as opposed to a change in mobility ?

2) Most of the figures show the decrease in intensity with T and show a non-linear decrease with an inflexion point. The authors have plotted the normalized intensity in a lin-lin plot. They could also have plotted the data in a log-log scale with different conclusions. Would the T_v be as clearly defined ? There is an ambiguity there

3) Another issue I have is the argument that the intrinsic decrease in intensity with temperature shown in figure S7 is irrelevant for the conclusions. I beg to disagree and suggest that the authors separate the intrinsic behavior from the AI behavior. In other words why not plot the normalized ratio of intensities for each temperature ? i.e the measured normalized decrease (figure S5) normalized by the intrinsic normalized decrease of the powder (figure S7a)

4) The most serious criticism is the following: the authors motivate their study with the need to predict creep behavior and define a "safe" usage region and a "recycling" region. A very worthwhile goal indeed. However the AIE that they detect is independent on the amount of catalyst while the creep behavior is not. How should one connect the T_v that they detect with the mechanical properties of the vitrimer then ?

5) The discussion on the Arrhenius dependence of the viscosity is also very unclear. Adding catalyst does not change the activation energy of the viscosity but it does change the prefactor A .

Corrections and changes made in response to the referee's comments

First of all, we are grateful to all referees for careful reading the manuscript and many useful suggestions that lead to essential improvements of our paper. We have tried our best to revise this article according to the comments of the referees.

(Referees' comments: in black; Corrections made by the authors in response to the comments: in red)

To Referee: 1

Comments:

This paper describes a new ingenious way to determine T_v in vitrimers. Vitrimers are the subject of intense researches both in basic science but also for practical applications. However, one of the limits in the application of vitrimers in real world, is the accurate measurement of the "working temperature" for them.

T_v , topology freezing transition temperature, represents the temperature limit above which the vitrimers cannot be used anymore for practical applications.

Authors propose to use AIE molecules as internal probes to determine T_v . As the fluorescence intensity of AIE molecules is sensitive to its environment (high intensity when the internal rotations of AIE are restricted, lower intensity when more freedom around AIE molecules increases the internal rotations of the AIE molecules), authors speculate that there will be possible to detect T_v by following the evolution of the fluorescence intensity with the T variations.

The technique works perfectly well. Authors tested the technique on different vitrimers to demonstrate the "universality" of the technique.

This paper, which describes a very original technique toward the accurate measurements of T_v in vitrimers, will have a great impact in this domain of research, which is a very hot subject. This paper is suitable for publication in Nature Communications. However, the English needs to be improved substantially.

P. Keller

Response: We are grateful for the very positive comments from the reviewer. According to the reviewer's suggestion, we have tried our best to improve the English in the revised version.

To Referee: 2

Comments:

This paper reported a sensitive fluorescence method for detection of topology freezing transition temperature (T_v). Luminogens with aggregation-induced-emission (AIE) characteristics were used as probes. The fluorescence intensity of polymer vitrimers doped with AIE luminogens changed sharply below and over T_v , which allowed detection of T_v . Moreover, it was found that T_v was independent of catalyst loading, by using the detection method. The method reported is novel and interesting.

Questions are listed as follows:

1. How does aggregation of dyes affect detection of T_v ? In epoxy vitrimers, TPE is homogeneously dispersed in polymer matrix. If dyes form aggregates in polymer matrices, does the method work well? Does inhomogeneous dispersion of dyes affect the results?

2. The authors chose the polymer vitrimers that are compatible with dyes. For the incompatible systems, does the method work well?

In summary, I recommend its publication after minor revision.

Response: We appreciate the referee's recognition on the novelty and significance of our work. We are very grateful for the referee's detailed advices.

Response to question 1: This is a good question. In our experiments, we did not observe the aggregation of AIE in the all cases. To further investigate the effect of aggregation, we added more experiments using the samples with a higher loading of TPE. According to XRD curves (Figure 1a here), even with a high TPE loading (3 wt% and 10 wt%), there is still no diffraction peak of TPE aggregation.

However, a high TPE loading does affect the measurement. For example, when doping 3 wt% TPE into epoxy vitrimer, T_v can also be clearly defined to $\sim 102^\circ\text{C}$, but a new transition at $\sim 165^\circ\text{C}$ (which may be derived from TPE) appears (Figure 1b). With 10 wt% TPE, more new transitions appear (Figure 1c). We actually do not fully understand how a higher TPE loading affects the measurement. Definitely this question needs to be addressed in more details, which will be done in our future work. In this work, we do not go through the detailed investigation as low concentration of TPE is enough.

In fact, one significant advantage of this method is that only a very small amount of AIE loading is needed, in which case AIE is homogeneously dispersed. And when doping with a higher TPE loading (10 wt% TPE), the sample is brittle. So low concentration of AIE not only means low-cost but also can avoid the possibility that a higher TPE loading may affect the thermal and mechanic properties of vitrimers.

Figure 1. (a) XRD curves of TPE-doped (containing 10 wt%, 3 wt%, 1 wt% and 0.1 wt% TPE) epoxy vitrimers and TPE powder. (b) The fluorescence intensity plot of 3 wt% TPE-doped epoxy vitrimer at a series of temperature at 470 nm (Excitation: 365 nm). All the fluorescence intensities are normalized to 45°C . (c) The fluorescence intensity plot of 10 wt% TPE-doped epoxy vitrimer at a series of temperature at 470 nm (Excitation: 365 nm). All the fluorescence intensities are normalized to 45°C .

Response to question 2: So far, the reported vitrimers are all based on common polymer matrixes. AIE molecules are organic molecules and have a very good compatibility with polymers. As there are many kinds of AIE molecules available, a wide selection to AIE molecules can be chosen to match various vitrimer systems. We think in case in the future there are very special vitrimers, which have poor compatibility with available AIE molecule, we may change the molecular structure of AIE correspondingly, so as to make AIE compatible to them.

To Referee: 3

Comments:

The authors apply the well-known effect of AIE to detect a change in fluorescence in doped vitrimers. The idea is ingenious and novel but the data interpretation in terms of topology-freezing transition is not sufficiently discussed in the paper, containing some serious methodological approximations that need to be better justified.

Response: We thank the referee for the positive comment. We are especially grateful for the questions and suggestions that the referee raised, which will lead to substantial improvement of the paper. We have very carefully gone through all these important points and addressed all of them in the revised version. The details are as follows:

1) A key point is to understand where does the change in fluorescence come from and what is the meaning of T_v based on what is known about AIE. Figure 1 is completely unclear. What difference should the reader see between above and below T_v ? Where is the decrease' in intensity coming from in a solid epoxy?

Response: Indeed Figure 1 was very unclear. We are sorry for such bad picture drawing ability. What we previously were going to show in Figure 1 is the following: below T_v , AIE molecules are in relatively rigid vitrimer matrixes. The intramolecular motions (i.e. rotation, vibration, torsion, and bending) of phenyl rings of AIE are restricted to some extent. The energy of the excited state is annihilated through radiation decay, and thus AIE emits efficiently. Above T_v , intramolecular motions of AIE are activated due to the movement of topology network, which is caused by the exchange reaction. Such movement significantly increases free volume in the vitrimer matrix. The intramolecular motions consume the energy of the excited state. This leads to the weak fluorescence emission of AIE. Therefore, a fluorescence intensity transition appears around T_v .

To make Figure 1 clearer, we changed to a new one (which is Figure 2 here) in the revised manuscript, and the corresponding caption has been changed to "AIE-doped vitrimer is highly emissive below T_v and weakly emissive above T_v ".

Figure 2. AIE-doped vitrimer is highly emissive below T_v and weakly emissive above T_v .

Are the luminogens becoming aggregated, i.e. insoluble in the epoxy. Is T_v the temperature where they become insoluble? If the luminogens can be infused in the epoxy with a solvent, they are presumably mobile

(the epoxy is above its T_g). Would the decrease in intensity not be due to a thermodynamically driven phase separation between luminogens and epoxy as opposed to a change in mobility?

Response: In our all experiments, we did not observe AIE luminogens became aggregated. As no other proper techniques are available to characterize the uniform dispersion of AIE molecules in polymers, XRD is the only method adopted so far by polymer researchers. We used the same technique. As indicated by the XRD data, there is no aggregation in our all systems. That is to say, the luminogens are soluble in the epoxy. Besides, the melting points of TPE is about 224°C. T_v we detected by this method is ~102°C. Therefore, T_v observed here is not the temperature where AIE luminogens become insoluble or aggregated.

The decrease in intensity is not due to phase separation either. The AIE luminogens are homogenous dispersed. If phase separation occurs, AIE luminogens would aggregate, which would increase the luminescence instead of weakening it.

2) Most of the figures show the decrease in intensity with T and show a non-linear decrease with an inflexion point. The authors have plotted the normalized intensity in a lin-lin plot. They could also have plotted the data in a log-log scale with different conclusions. Would the T_v be as clearly defined? There is an ambiguity there.

Response: It is a very good idea. T_v could also be detected in a log-log scale plot. For example, we plotted the data of epoxy vitrimer doped with 1 wt% TPE in a log-log scale (Figure 3), T_v could also be clearly defined to ~102°C, which is almost the same as that in a lin-lin plot.

Figure 3. The fluorescence intensity plot in a log-log scale of 1 wt% TPE-doped epoxy vitrimer at a series of temperature at 470 nm (Excitation: 365 nm). All the fluorescence intensities are normalized to 45°C.

3) Another issue I have is the argument that the intrinsic decrease in intensity with temperature shown in figure S7 is irrelevant for the conclusions. I beg to disagree and suggest that the authors separate the intrinsic behavior from the AIE behavior. In other words why not plot the normalized ratio of intensities for each temperature? i.e the measured normalized decrease (figure S5) normalized by the intrinsic normalized decrease of the powder (figure S7a)

Response: Yes, the referee is right, the intrinsic decrease in intensity with temperature of AIE powder is irrelevant for the conclusions. We are very sorry for the previous justification. The referee's suggestion is very helpful! It is best to plot the normalized ratio of intensities for each temperature. However, when we did this, we found that it was very hard. AIE powder is highly aggregated, which emits extremely strong fluorescence and much stronger than that of AIE-doped vitrimer. The intensity decrease behaviour of the aggregated AIE powder is completely different from a AIE molecule. So we cannot normalize the measured normalized decrease (figure S5) by the intrinsic normalized decrease of the powder (figure S7a). However,

according to previous research, the intrinsic behaviour of the AIE has little influence on the determination of polymer chain mobility. When AIE-probe technique was used to determine T_g of polymers (such as polystyrene (PS, T_g is 80°C), and poly(methyl methacrylate) (PMMA, T_g is 103°C)) (*Polym. Chem.* 2015, 6, 3537), there is just one turning point of T_g in the plot of intensity decrease with temperature if AIE is homogeneously dispersed. That is to say, there is no turning point derived from AIE itself. Therefore, the transition is not caused by TPE itself, but by the appearance of T_v . Without normalization, the data present in this manuscript is still valid.

We deleted Figure S7 in the revised version, and added corresponding description in the main text, which is “When AIE-probe technique was used to determine T_g of polymers (such as polystyrene (PS, T_g is 80°C), and poly(methyl methacrylate) (PMMA, T_g is 103°C)), there is just one turning point of T_g in the plot of intensity decrease with temperature if AIE is homogeneously dispersed. That is to say, there is no turning point derived from AIE itself. Therefore, the transition is not caused by TPE itself, but by the appearance of T_v ”.

- 4) The most serious criticism is the following: the authors motivate their study with the need to predict creep behavior and define a “safe” usage region and a “recycling” region. A very worthwhile goal indeed. However the AIE that they detect is independent on the amount of catalyst while the creep behavior is not. How should one connect the T_v that they detect with the mechanical properties of the vitrimer then?
- 5) The discussion on the Arrhenius dependence of the viscosity is also very unclear. Adding catalyst does not change the activation energy of the viscosity but it does change the prefactor A.

Response: Those two questions are in fact related to each other. So we answer them together here.

All polymers have their characteristic temperatures such as glass transition temperature (T_g), melting point (T_m), and so on. Those temperatures provide an indication on the “safe” utilization temperature limits. For vitrimers, we think that there should be another characteristic temperature, which is T_v , besides those common ones. DSC is the widely used technique to determine T_g and T_m , which are measured in a static condition. T_m and T_g change when the material is under external force. However, those measured T_g and T_m are very important for safe utilization temperature determination. The situation of T_v here is very similar. T_v measured by the AIE method here is exactly similar to the T_g and T_m measured by DSC.

T_v is independent on the amount of catalyst while the creep behavior is not. This seems very confusing, but it is not contradictory. We are sorry that we failed to explain this clearly in the previous manuscript. The creep is depended on the catalyst because catalyst can accelerate the reaction rate. However, catalyst cannot change the intrinsic T_v of the network. As we can deduce from the following Arrhenius equation:

$$k = Ae^{\frac{-E_a}{RT}} \quad (1)$$

$$\ln k = \ln A - \frac{E_a}{RT} \quad (2)$$

$$\frac{d \ln k}{dT} = \frac{E_a}{RT^2} \quad (3)$$

From previous studies, it has been proved that E_a is unchanged when catalyst loading changes. The k is small below T_v and large above T_v . So, from its differential form in equation (3), the exchangeable reaction rate or viscosity ($\frac{d \ln k}{dT}$), which can be negatively reflected here by the fluorescence intensity plot as a function of increasing temperature precisely, is only connected with temperature, as E_a is unchanged and only T is a

variable in $\frac{d \ln k}{dT}$. That is to say, T_v , signifying the exchangeable reaction rate ($\frac{d \ln k}{dT}$), is only connected with temperature and is not related to catalyst loading.

The referee is right, adding catalyst does change the prefactor A. But catalyst accelerates the reaction at all temperatures. When taking a derivative of equation (2) with respect to temperature, $d(\ln A)$ is zero. It does not affect the change of slope rate (which is used for identifying T_v) in fluorescence intensity-temperature plot.

Like T_g and T_m measured by DSC, the T_v here can be an additional characteristic temperature of vitrimers. It can be regarded as the upper limit temperature for utilization and the lower limit temperature for processing. Similar to the common polymers with known T_g and T_m , when it is necessary to determine the exact safe temperature, more characterizations on creep and mechanical properties are needed, as they are influenced by various factors.

Thanks to the referee, we realized that we did not make this clear to readers. So, we added the corresponding explanation in the revised manuscript.

REVIEWERS' COMMENTS:

Reviewer #2 (Remarks to the Author):

The authors have addressed all my concerns. I recommend its publication now.

Reviewer #3 (Remarks to the Author):

The authors have addressed thoughtfully and correctly the comments of the reviewer and clarified some of the points that were a bit obscure.

Point-by-point response to the referee's comments

We are grateful to two referees for careful reading the manuscript again. We also appreciate two referees' recognition of our manuscript.

REVIEWERS' COMMENTS:

Reviewer #2:

The authors have addressed all my concerns. I recommend its publication now.

Reviewer #3:

The authors have addressed thoughtfully and correctly the comments of the reviewer and clarified some of the points that were a bit obscure.